# Enhancing colorectal polyp classification using gaze-based attention networks

Zhenghao Guo[1,*], Yanyan Hu[2,*], Peixuan Ge[3], In Neng Chan[3], Tao Yan[1,2,3], Pak Kin Wong[3], Shaoyong Xu[2], Zheng Li[2] and Shan Gao[2]

[1] School of Mechanical Engineering, Hubei University of Arts and Science, Xiangyang, China
[2] Xiangyang Central Hospital, Affiliated Hospital of Hubei University of Arts and Science, Xiangyang, China
[3] Department of Electromechanical Engineering, University of Macau, Taipa, Macao, China
* These authors contributed equally to this work.



## ABSTRACT

Colorectal polyps are potential precursor lesions of colorectal cancer. Accurate classification of colorectal polyps during endoscopy is crucial for early diagnosis and effective treatment. Automatic and accurate classification of colorectal polyps based on convolutional neural networks (CNNs) during endoscopy is vital for assisting endoscopists in diagnosis and treatment. However, this task remains challenging due to difficulties in the data acquisition and annotation processes, the poor interpretability of the data output, and the lack of widespread acceptance of the CNN models by clinicians. This study proposes an innovative approach that utilizes gaze attention information from endoscopists as an auxiliary supervisory signal to train a CNN-based model for the classification of colorectal polyps. Gaze information from the reading of endoscopic images was first recorded through an eye-tracker. Then, the gaze information was processed and applied to supervise the CNN model's attention via an *attention consistency module*. Comprehensive experiments were conducted on a dataset that contained three types of colorectal polyps. The results showed that EfficientNet_b1 with supervised gaze information achieved an overall test accuracy of 86.96%, a precision of 87.92%, a recall of 88.41%, an F1 score of 88.16%, the area under the receiver operating characteristic (ROC) curve (AUC) is 0.9022. All evaluation metrics surpassed those of EfficientNet_b1 without gaze information supervision. The class activation maps generated by the proposed network also indicate that the endoscopist's gaze-attention information, as auxiliary prior knowledge, increases the accuracy of colorectal polyp classification, offering a new solution to the field of medical image analysis.

## INTRODUCTION

According to worldwide cancer statistics, colorectal cancer (CRC) ranks as the second leading cause of cancer-related deaths and the third most prevalent cancer type. Annually, this disease accounts for over 1.85 million new cancer cases (9.8% of total cancer diagnoses) and results in approximately 850,000 cancer deaths (9.2% of total cancer-related deaths), highlighting CRC as a notable public health hazard and a major challenge for global medical systems (*Sung et al., 2021*; *Siegel et al., 2023*).

Corresponding authors
Tao Yan, yantao@hbuas.edu.cn
Pak Kin Wong, fstpkw@um.edu.mo

In clinical practice, the screening process for CRC mainly involves detecting colorectal polyps through endoscopic examination and then performing biopsies on these polyps to determine the nature of the lesion (*Wang & Dong, 2020*). However, the effectiveness of CRC screening is influenced by several factors, including the endoscopist's operating skills, the patient's preoperative preparation, and the visibility of colorectal polyps. These factors can significantly affect the quality of screening results and lead to a high rate of underdiagnosis and misdiagnosis of colorectal polyps. The missed diagnosis rates of early CRC range from 2% to 26% (*Zhao et al., 2019*; *Leufkens et al., 2012*). Therefore, more precise detection and more accurate classification of CRC could be greatly beneficial to clinical practice.

In recent years, convolutional neural networks (CNNs) have been widely adopted in the field of endoscopy due to their ability to automatically extract features from endoscopic images (*Yan, Wong & Qin, 2021*). The robust learning capability of CNNs not only enhances the accuracy of colorectal polyp classification but also greatly reduces the workload of endoscopists. Additionally, the objectivity and high efficiency of the CNN-based systems can reduce the rate of misdiagnosis and underdiagnosis of colorectal polyps (*Pamudurthy, Lodhia & Konda, 2019*). With the advancement of CNN models, the rate of lesion misdiagnosis and underdiagnosis in colonoscopy has been effectively reduced, offering insights for future automated endoscopic diagnosis and treatment.

Screening colonoscopies are essential for the prevention of CRC. With the development of CNNs, real-time computer-aided detection for CRC is now available in clinical practices (*Saito et al., 2020*). These CNN-based systems have improved the detection rate of colorectal polyps, including differentiating between precancerous growths and noncancerous hyperplastic polyps. Endoscopists can spend substantial time searching for small polyps, which are often hyperplastic. Since malignant transformation of these polyps is rare, pathologic evaluation may be unnecessary, and removal of these polyps increases healthcare costs. Some patients with early CRC that was diagnosed after surgical resection may be treated with minimally invasive endoscopic therapy if their histopathologic findings are known before surgery. Therefore, accurate histologic diagnosis before endoscopic resection can potentially prevent unnecessary endoscopic treatment and significantly reduce the financial burden (*Chen et al., 2018*).

The Narrow Band Imaging International Colorectal Endoscopic (NICE) classification is a simplified system for the diagnosis of colorectal polyp histology that is widely used in centers without magnifying endoscopic capabilities. The NICE classification system divides colorectal polyps into three types (*Hewett et al., 2012*). Type 1 includes hyperplastic and inflammatory polyps, which are usually benign and are unlikely to develop into cancer. Type 2 consists of adenomatous polyps, intramucosal carcinoma, and superficial submucosal invasive carcinoma, which have a high potential risk of becoming cancerous. Type 3 is deep submucosal invasive carcinoma, which is the highest risk type as it indicates that the tumor has penetrated into the submucosal layer or deeper tissues. Although NICE classification has made some progress in accurately identifying polyp types, the consistency of this accuracy can vary depending on the case, especially when observations

are conducted by non-experts (*Ladabaum et al., 2013*). The CNN-based colorectal polyp pathology prediction can help overcome this challenge.

Although CNN-based models have made significant progress in detecting gastrointestinal diseases, training deep CNN models remains challenging due to difficulties in collecting medical data, such as privacy concerns, the need for expert annotations, data imbalance, and the scarcity of high-quality labeled datasets. Additionally, engaging in meticulous annotation tasks is time-consuming and labor-intensive, amplifying the workload for clinical experts. Previous studies often focused on improving the accuracy of CNN models, but these models often lack interpretability and may not be accepted by clinical professionals in the medical field (*Zhang & Zhu, 2018*). Recently, researchers have started integrating doctors' eye gaze information into diagnostic analyses to address these limitations of CNN-based methods (*Bisogni et al., 2024*; *Ibragimov & Mello-Thoms, 2024*).

Although eye gaze has been widely used in many research fields, its integration into CNN-based systems is still in its infancy. The use of eye gaze information in automatic auxiliary diagnosis has great value and potential. By using eye-tracking technology, it is possible to create a new medical image annotation method that is comparable to traditional manual annotation (*Bhattacharya, Jain & Prasanna, 2022*). Eye gaze data depicts the search pattern of doctors when looking for tumors or suspicious lesions in scans, revealing the locations where doctors tend to linger during diagnostic screening. These areas are important because prolonged gaze often indicates regions that may contain abnormalities or potential lesions, thus providing valuable additional information for diagnosis. This additional high-level information can guide CNN models to learn disease features in an interpretable way. Therefore, embedding eye gaze information into diagnostic analysis has become a trending topic in recent years (*Ibragimov & Mello-Thoms, 2024*; *Neves et al., 2024*).

This study proposes an innovative attention network based on gaze, which utilizes the gaze information of endoscopists to improve the accuracy and interpretability of colorectal polyp classification while reducing the amount of training image data. Gaze information is captured via an eye-tracker, which can be integrated into the endoscopists' image interpretation process without interference. The gaze information acquired during endoscopic image reading is considered a priori medical knowledge that is used as an auxiliary supervisory signal to improve the training of the CNNs. This approach not only improves the performance of the model for identifying rare diseases but also reduces the cost of the annotation process.

The main contributions of this study are summarized as follows: (1) The construction and publication of a gaze dataset for colorectal polyps that includes images of colorectal polyps and the corresponding gaze maps; (2) the design of a gaze-based attention network for the classification of colorectal polyps that can enhance accuracy and robustness by using eye gaze information as a supervised source; (3) the comprehensive analysis and comparison of different CNN backbones incorporating gaze attention information in order to verify the effectiveness of the gaze attention mechanism in classifying colorectal polyps.

The rest of this article is organized as follows: 'Related Work' briefly reviews the literature related to this study, including deep learning based colorectal polyp classification methods and the application of visual attention in medical image analysis; 'Methods' provides a detailed description of the deep learning network based on gaze attention proposed in this article, including the design of gaze data collection, the attention consistency module, and the classification network; 'Experiments' introduces the construction of the dataset, experimental setup, and evaluation indicators; 'Results' presents the experimental results and verifies the effectiveness of the method through comparative analysis; the 'Conclusion' summarizes the main contributions and research limitations of this article and explores future research directions.

## RELATED WORK

This section examines recent research advances regarding the classification of colorectal polyps and presents a detailed review of the relevant research focusing on gaze attention for medical image analysis.

### Classification of colorectal polyps based on CNNs

The classification of early colorectal polyps mainly relies on manual identification by endoscopists. This method is very time-consuming and cumbersome for physicians, and it also creates difficulties in ensuring the accuracy of classification due to the subjectivity of manual identification. The need for endoscopists to carefully observe and judge a large quantity of images can easily cause fatigue and affect the accuracy of diagnosis. In traditional research methods, researchers often use feature extraction technology to collect information such as shape, texture, and color from polyp images. These features can provide information for identifying and classifying different types of polyps. However, the feature extraction process is complex and requires researchers to have considerable professional knowledge and experience. Additionally, the extracted features must be further processed and screened to ensure their effectiveness for performing the classification task (*Patel et al., 2020*). With the improved performance of deep neural networks, CNNs are increasingly used in the classification of colorectal polyps. Compared with traditional manual feature extraction methods, CNNs can more efficiently extract abstract and high-level features, thereby improving the accuracy of classification. A CNN-based system can automatically process and analyze a high volume of endoscopic images, significantly reducing the doctor's review time. In addition, CNN-based methods reduce the risk of misdiagnosis and underdiagnosis, providing patients with more timely and accurate treatment options (*Sánchez-Peralta et al., 2020*).

Various advanced CNNs have emerged to aid in the classification of colorectal polyps. *Liu et al. (2023)* proposed Polyp DeNet, which combines dilated convolution and efficient channel attention modules based on ResNet-50 and uses parameter sharing and data augmentation methods to improve model performance. This model has achieved a high level of accuracy in colorectal polyp classification, fully demonstrating the benefit of artificial intelligence-assisted diagnosis. *Sharma et al. (2023)* proposed a unique lightweight CNN with a discrete wavelet pooling strategy for colon polyp classification.

This model uses wavelet pooling instead of ordinary pooling layers to achieve a better balance between receptive field size and computational efficiency. *Rahman, Wadud & Hasan (2021)* used an integrated model combining Xception, ResNet-101, and VGG-19 networks to achieve higher accuracy in classifying hyperplastic, serrated, and adenomatous polyps, as well as non-polyps. *Tanwar et al. (2022)* proposed a single-shot multiframe detector (SSD) architecture with image preprocessing and added dropout in the fully connected layers to prevent overfitting and improve the generalization ability of the model. This assisted in effectively detecting and classifying colorectal polyps in colonoscopy images. *Hossain et al. (2023)* proposed an autonomous CRC screening method to detect polyps and assess their potential threat. In that study, DoubleU-Net was used for polyp segmentation, and Vision Transformer (ViT) was used for classification according to polyp risk. The proposed method classified polyps in the Endotech Challenge and Kvarsi-SEG datasets as either hyperplasic or adenomatous with a test accuracy of 99%.

Although deep learning models play an important role in colorectal polyp classification, and CNN-based diagnostic models have shown excellent performance, they still have some limitations. CNNs are data-driven algorithms that require a large amount of labeled data. The annotation process of datasets is very cumbersome, labor-intensive, and time-consuming, posing a challenge to doctors. Additionally, these models have poor interpretability, and doctors require more information to guide their decision-making process. In a medical setting, these models must maintain high accuracy and provide clear explanations of the predicted results.

## Eye-tracking-based medical image analysis

Eye-tracking technology provides a potential solution to the problems of cumbersome annotation and insufficient interpretability of CNNs. By tracking the eye movements of doctors while they view endoscopic images, this technology can provide valuable data for developing more interpretable and transparent classification models. These data help reveal key features and areas in colorectal polyp images, thereby improving the interpretability of the model. Therefore, combining eye-tracking technology with CNNs may be a promising way to improve colorectal polyp classification.

In recent years, eye-tracking technology has been widely applied in the medical field as auxiliary prior knowledge to assist in training more efficient CNN models. *Stember et al. (2019)* proposed a dynamic annotation method based on eye-tracking for lesions and organs in multimodal medical images including computed tomography (CT) and magnetic resonance imaging (MRI) and applied eye-tracking information to automatic image annotation. *Karargyris et al. (2021)* proposed a multi-head model based on U-Net that incorporates radiologists' gaze maps for chest X-rays. The clinical-expert gaze maps generated by eye-tracking equipment can be used as an alternative to weakly supervised labels to guide the training of deep learning models. This method was used to simultaneously predict the expert's attention patterns and their classification results to distinguish between congestive heart failure, pneumonia, and healthy tissue. *Wang et al. (2022)* proposed a gaze-guided attention model based on X-ray images to classify osteoarthritis into four categories with good interpretability and classification

**Table 1 Articles related to gaze attention.**

| Ref. | Year | Imaging | Study design | Study aim | DL model | Dataset | Outcomes |
|---|---|---|---|---|---|---|---|
| Stember et al. (2019) | 2019 | MRI | Retrospective | Addressing the issue of lack of annotated data through eye tracking technology | __ | 356 images | Eye tracking can provide an efficient way to generate annotated data for training deep learning segmentation models, without the need for specialized manual annotation |
| Karargyris et al. (2021) | 2021 | CXR | Retrospective | Research on how to use eye gaze data to explain the prediction results of the model | U-Net | 1,083 images | Provides an inexpensive and efficient way to approximate annotated images of regions of interest by using the radiologist's eye gaze |
| Wang et al. (2022) | 2022 | X-ray | Retrospective | Demonstrate the eye movement of radiologists reading medical images can be a new form of supervision to train the DNN-based computer-aided diagnosis system | ResNet | 2,000 images | Integrate radiologist eye data into CAD systems to enhance their performance. |
| Wang et al. (2024) | 2024 | X-ray | Retrospective | The gaze information collected by eye-tracking devices is converted into visual attention maps, which is a time-consuming preprocessing step that ihnders the daily work of radiologists | __ | 1,083 images | Introducing eye gaze data can significantly improve the classification performance and anomaly localization ability of the model, with an experimental accuracy of 83.18% |
| Ma et al. (2023) | 2023 | Color image | Retrospective | Improving automated diagnosis by using gaze heatmaps as network output | ViT | 8,367 images | And proposed a new saliency guided visual transformer that suppresses and corrects rapid learning by injecting artificial prior knowledge. |
| Wang, Zhang & Ge, 2023 | 2023 | CT | Retrospective | Using eye movement information from doctors as additional supervision to improve the accuracy of abdominal organ segmentation | __ | 4,230 images | Using gaze attention as an auxiliary supervision mechanism for network training, the segmentation accuracy reached 81.87% (Dice similarity coefficient) and 11.96% (Hausdorff distance) |
| Jiang et al. (2024) | 2024 | Fundus image | Retrospective | Early diabetes retinopathy brings challenges to clinical diagnosis, leading to limited research in this field | SAM | 394 images | Adding eye movement information from doctors can improve the annotation efficiency of clinical doctors and enhance segmentation performance through fine-tuning using annotations |

Note:
DL, Deep learning; MRI, Magnetic resonance imaging; CXR, Chest X-Ray; CT, Computed tomography; ViT, Vision transformer; SAM, Segment anything model.

performance. Another study (Wang et al., 2024) proposed a gaze-guided graph neural network (GNN). As a real-time, realistic, end-to-end disease classification algorithm, GNN integrates raw eye gaze data and can achieve high-accuracy classification without preprocessing. Ma et al. (2023) proposed a novel and effective saliency-guided visual transformer (SGT) model that utilizes eye movement data to correct ViT shortcut learning. This model can effectively learn and use human prior knowledge, perform well in multiple datasets, and improve the interpretability of the ViT model. Wang, Zhang & Ge (2023) proposed a method that combines the gaze information of radiologists with image labeling requirements. A dual-path encoder is used to integrate gaze information, and a cross-attention transformer module is used to embed the gaze patterns of doctors reading images into the network model. Through multi-feature skip connections, spatial information is

combined in the downsampling process to offset the internal details of the segmentation, achieving the high-precision segmentation requirements. *Jiang et al. (2024)* proposed an eye-tracking-based early diabetic retinopathy (DR) detection model using ophthalmologists' gaze maps. The weighted gaze map is integrated as a supervision mask to guide the learning of the DNN model's attention. A novel difficulty-aware and class-adaptive gaze-map attention learning strategy was proposed to enhance the interpretability of the model. The attention-guided approach regularized by the Class Activation Map (CAM; *Zhou et al., 2016*) shows improvements in the accuracy and interpretability of early DR detection models. The key information of these studies is summarized in Table 1.

These studies highlight the potential of eye-tracking technology in the field of medical imaging diagnosis. Eye-tracking technology not only acts as auxiliary prior knowledge to aid in training deep learning models but also enhances the interpretability and comprehensibility of these models. By integrating eye-tracking data, researchers gain deeper insights into the attention and decision-making processes of medical experts, thereby refining the training strategies and performance of deep learning models. Moreover, eye-tracking technology can serve as an alternative weakly supervised label, guiding the training of deep learning models and thereby enhancing their accuracy and credibility in medical imaging diagnosis.

# METHODS

This study was approved by the Institutional Review Board of Xiangyang Central Hospital (IRB approval number: 2024-145). Since this was a retrospective study, the Institutional Review Board waived the written informed consent requirement.

This study introduces a gaze-based attention network for classifying colorectal polyps based on the fixation images of clinical doctors during the diagnostic process. First, diagnostic gaze information (*i.e.,* medical prior knowledge) from clinical doctors was obtained and processed. Then, the classification network for colorectal polyps was constructed using the processed gaze maps.

## Eye-tracking data collection process

Standardized data collection procedures were used for the collection of high-quality eye-tracking data. The main body of the eye-tracking device used in this study included a 1,920 × 1,080 27-inch high-definition liquid crystal display (LCD) and a Tobii Eye Tracker 5 eye-tracking device. The eye-tracking device connected to an external computer via a data cable and was used to record binocular gaze data.

This study implemented a customized data collection software in Python (*Wang et al., 2022*) that quickly constructed the experimental environment through simple parameter settings. Eye-tracking data was collected from endoscopists at a frequency of 90 Hz. Endoscopists sat on a fixed chair and kept their eyes perpendicular to the screen before providing eye movement data that simulated clinical working conditions. Following the user manual of the Tobii Eye Tracker 5 eye-tracking device, the distance between the endoscopists' eyes and the screen was adjusted within the range of 50 to 80 cm. The eye-tracking device calibration was personalized for each doctor to account for different

reading habits, which helped adapt to the vision of different individuals and reduce eye-tracking errors. Each clinician performed a nine-point calibration procedure during first use of the eye-tracking device.

During the diagnosis of colorectal polyps, each image was displayed in the center of the LCD screen, and the endoscopist entered their diagnosis using the numeric keys on the keyboard. For example, pressing the number "1" indicated that the polyp type was diagnosed as Type 1, and "Enter" indicated that the diagnosis was completed, prompting the next image. During this process, the eye-tracking device continuously recorded the participant's eye movement information including the saccade point, fixation point, fixation duration, and eye scanning path, all of which was stored in the computer.

## Gaze attention heatmap generation

During the eye-tracking data collection process, a series of gaze points on each diagnostic image was recorded in real-time. However, despite personalized calibration, slight deviations in the endoscopists' gaze points could still occur due to the inherent system error of the equipment. To reduce these deviations, a Gaussian function $G(x, y)$ (shown as Eq. (1)) was applied to transform each gaze point into the gaze area.

$$G(x, y) = \frac{1}{\sqrt{2\pi}\sigma} e^{\frac{-[(x-x_c)^2 + (y-y_c)^2]}{2\sigma^2}}. \tag{1}$$

In this equation, $(x_c, y_c)$ represented the central gaze point, and the pixel level distance of variance $\sigma$ was used as the effective field of view range. In this study, a higher $\sigma$ represented a broader range of focus for the endoscopists' attention, potentially encompassing areas unrelated to the diagnosis. However, a lower $\sigma$ could impede the ability of the clinician to concentrate on a specific lesion area, resulting in attention being divided among several locations inside a single lesion and raising computational requirements. It is appropriate to use a value of $\sigma$ that modestly compensates for human visual error. The following proportion equation was developed:

$$\frac{\pi(\sigma_1)^2}{H \cdot W} = \frac{\pi\sigma^2}{H_p \cdot W_p}. \tag{2}$$

$H_P = 1{,}080$, $W_p = 1{,}920$ represented the display resolution, and $H = 48$ and $W = 64$ were the screen physical values corresponding to $H_P$ and $W_p$, respectively. According to geometric operations, $\sigma_1 = \pi R\theta/360°$, in which $\theta$ represents perspective error. The pixel variance $\sigma$ on the display was estimated as:

$$\sigma = \frac{\theta}{360°} \pi R \sqrt{\frac{H_p \cdot W_p}{H \cdot W}}. \tag{3}$$

In this equation, $R$ is the approximate distance area of 50 to 80 cm between the eyeball and the fixation point on the screen. Inspired by *Jiang et al. (2024)*, $\theta$ was set to $1°$ to offset most of the errors. As specified in the user manual for the Tobii Eye Tracker 5, the maximum angular deviation for a gaze point was approximately $\pm 0.5°$. Corresponding to

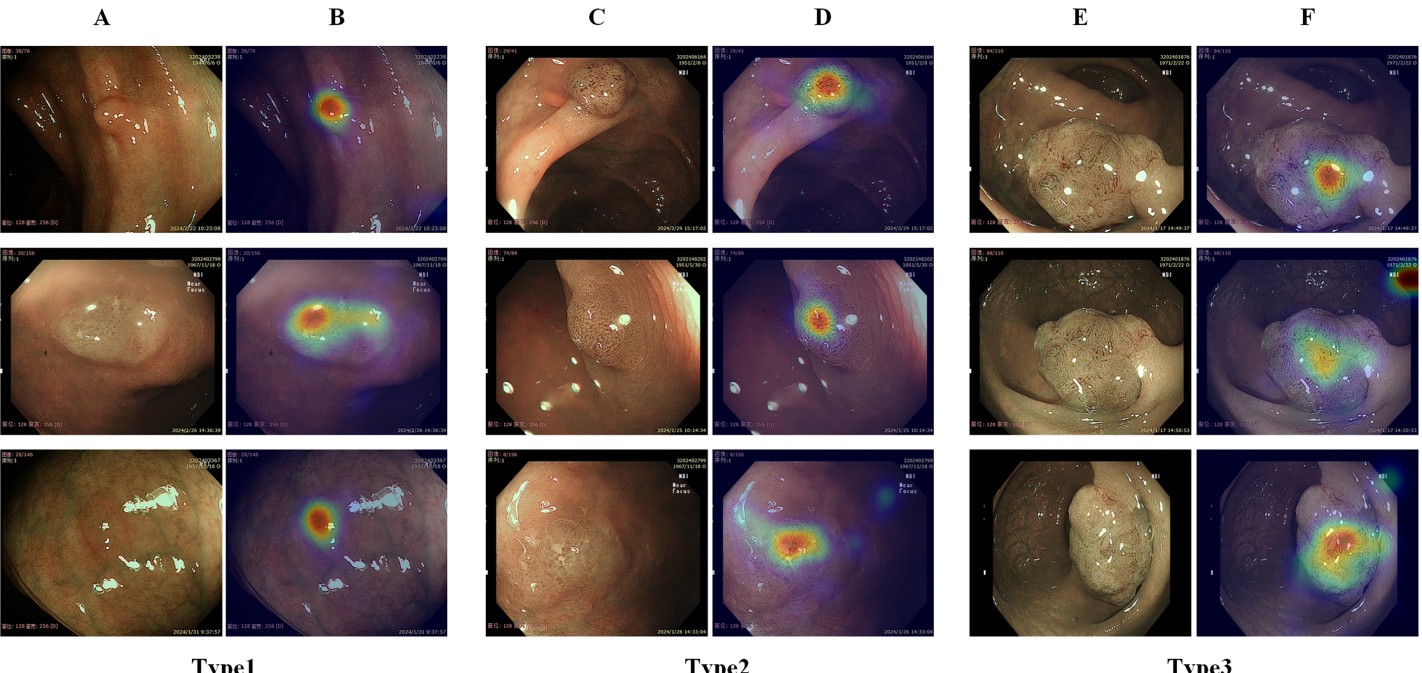

**Figure 1 Illustration of different types of colorectal polyps and corresponding gaze attention heatmaps.** (A) Diagram of a hyperplastic or inflammatory polyp; (B) gaze attention heatmaps of a hyperplastic or inflammatory polyp; (C) diagram of an adenomatous polyp, intramucosal carcinoma, and superficial submucosal invasive carcinoma; (D) gaze attention heatmaps of an adenomatous polyp, intramucosal carcinoma, and superficial submucosal invasive carcinoma; (E) diagram of a deep submucosal invasive carcinoma; (F) gaze attention heatmaps of a deep submucosal invasive carcinoma.

the interval $[-0.5°, 0.5°]$ Finally, the determined $\sigma$ value fell between 17.1 ($R = 50$ cm) and 27.2 ($R = 80$ cm). As a result, $\sigma$ was pre-set to 30 to better mitigate the errors.

The GaussianBlur function in the OpenCV library was used to perform Gaussian blur processing to reduce the noise and details in the image. A Gaussian kernel of size 199 was applied to the input image and the variance was set to 30. This process can effectively smooth the image by weighted-averaging each pixel and its surrounding neighborhood pixels, making it Smoother and effectively reducing the raw data of the eye-tracker in the image, including information that is irrelevant to the task.

After this process, the endoscopists' gaze attention maps were generated, and these maps were then superimposed onto the original endoscopic images to generate the gaze attention heatmaps (Fig. 1), which could be applied to supervise the network attention heatmaps via an attention consistency loss.

### Gaze-based attention network

The proposed gaze-based attention network aimed to align network attention with external supervision of the visual attention of endoscopists. This study used EfficientNet_b1, which achieved the best performance in the following experiments Compared with other general models, as the backbone to construct the gaze-based attention network. The architecture of EfficientNet_b1 is shown in Fig. 2. In EfficientNet_b1, the mobile inverted bottleneck convolution (MBConv) block served as a

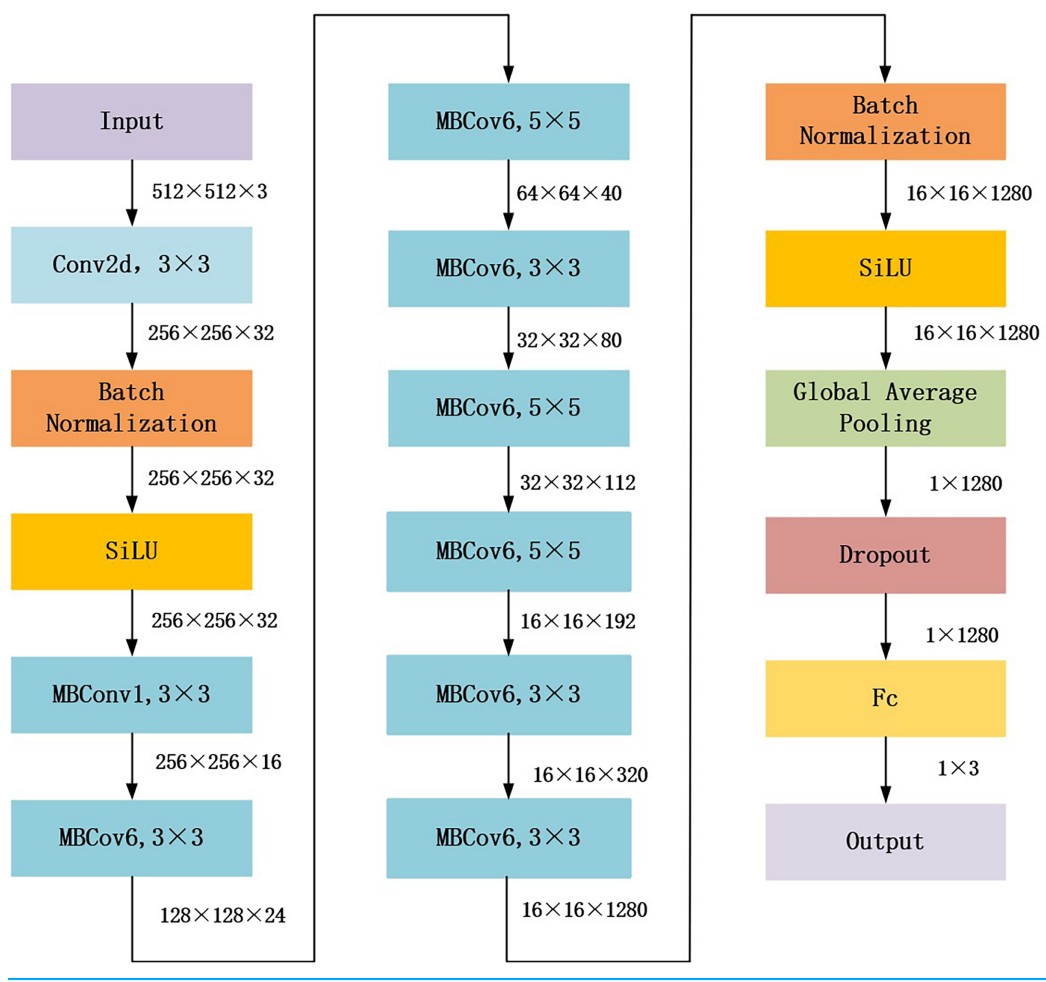

**Figure 2 Architecture of EfficientNet_b1 backbone.**

special residual module; it adopted lightweight depth-wise separable convolution and squeeze and extraction (SE) modules to improve the efficiency and performance of the network. The core components of MBConv blocks are divided into two stages: deep convolution and pointwise convolution (*Tan & Le, 2019*). In the deep convolution stage, the MBConv block independently processes the features of each channel rather than all channels simultaneously, which helps to extract more refined feature information. The output feature map is convolved with a 1 × 1 convolution kernel in the pointwise convolution stage to adjust the number of channels to meet the desired value. Additionally, the SE module in the MBConv block dynamically adjusts the importance between channels, enhancing feature representation. These advantages ensure that the proposed gaze-based attention network has strong feature extraction capabilities.

The proposed gaze-based attention network, as shown in Fig. 3, contained a classification module and an attention consistency module. In the classification module, the input image was first processed through the EfficientNet_b1 backbone to obtain a 16 × 16 × 1,280 feature map. Then, global average pooling (GAP) and flatten applications were applied to obtain a 1 × 1 × 1,280 feature vector. Finally, a fully connected layer of three

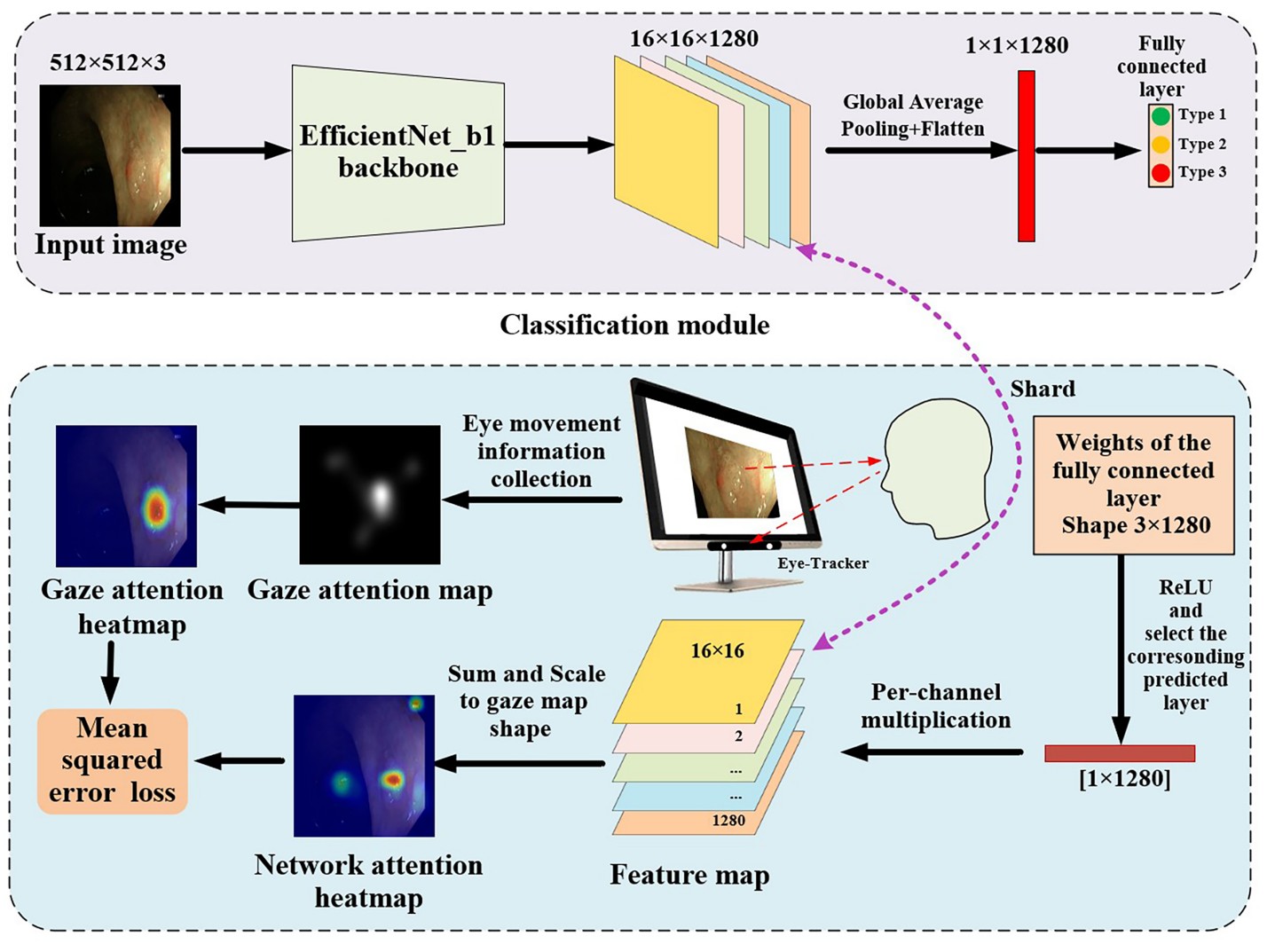

**Figure 3 Architecture of gaze-based attention network for colorectal polyp classification.**

nodes (representing the three categories of polyps) was used to obtain the final classification result.

In the attention consistency module, weights from the fully connected layer were taken to generate the model feature map. After applying the ReLU function, only the set of weights of the predicted class, which had a $1 \times 1{,}280$ shape, was used to perform channel-wise multiplication with the $16 \times 16 \times 1{,}280$ feature map, and the maps along channels were summed to get a single $16 \times 16$ map. This map was scaled to the same shape as the gaze attention map and was activated through the sigmoid function to acquire the network attention heatmap. To achieve multi-task learning, this study designed a joint loss function that included classification loss and regression loss to optimize the performance of the model on both classification and regression tasks. In classification tasks, cross-entropy loss

(CE) was used to optimize the classification ability of the model to more closely match the true labels.

CE was used to measure the difference between the predicted category (x) and the true category (label) of the model. The CE loss can be formulated as:

$$CE\_Loss = -\sum_{i=1}^{n} y_i \log(p_i). \tag{4}$$

In this equation, n represents the number of classifications, $y_i$ represents the true label of the i-th class, and $p_i$ represents the predicted label of the i-th class by the model.

In the regression task, the model used gaze attention heatmaps and network attention heatmaps to calculate the mean squared error (MSE) loss function. Mean square error measures the accuracy of model prediction by calculating the square difference between the predicted and actual values of each sample and then averaging the differences among all samples.

Using MSE as the regression loss function to measure the difference between two attention maps, the MSE loss can be formulated as:

$$MSE\_Loss = \frac{1}{N}\sum_{i=1}^{N}\sum_{j=1}^{M} (A_{model}(i,j) - A_{human}(i,j))^2. \tag{5}$$

In this equation, N and M are the height and width of the image, respectively, while $A_{model}(i,j)$ and $A_{human}(i,j)$ are the values of the network attention heatmap and the endoscopists' eye gaze attention heatmap, respectively, at position (i,j). This loss function represents the difference at each pixel position between the attention heatmap generated by the model and the endoscopists' eye gaze attention heatmap and calculates the loss by summing up the differences at all positions.

The total loss combines the classification task (CE_Loss) and the gaze prediction task (MSE_Loss). By using weighted combination, this value can provide insight into the quality of the classification and show that the predicted heat map of the network is close to the true value. The total loss can be formulated as:

$$LOSS = CE\_Loss + MSE\_Loss. \tag{6}$$

## EXPERIMENTS

### Dataset construction

The colorectal polyp gaze dataset was constructed using Narrow Band Imaging (NBI) images of colorectal polyps and the corresponding gaze attention images (*i.e.,* gaze attention maps and gaze attention heatmaps). All data collection and annotation processes were carried out by following the tenets of the Declaration of Helsinki.

A retrospective search was conducted of the NBI colonoscopy observation data at Xiangyang Central Hospital from January 1, 2023 to March 10, 2024. The criterion for inclusion was that patients with colorectal polyps must also have had corresponding

**Table 2 Colorectal polyps gaze dataset.**

| NICE classification | Training set (~60%) | | Validation set (~20%) | | Test-set (~20%) | |
|---|---|---|---|---|---|---|
| | No. of Lesions | No. of images | No. of Lesions | No. of images | No. of Lesions | No. of images |
| Type 1 | 34 | 94 | 30 | 30 | 28 | 28 |
| Type 2 | 61 | 251 | 12 | 78 | 58 | 82 |
| Type 3 | 3 | 13 | 2 | 3 | 2 | 5 |

Notes:
Type 1: proliferative or inflammatory polyps; Type 2: adenomatous polyps, intramucosal or superficial submucosal infiltrating carcinoma; Type 3: deep submucosal infiltrating carcinoma.

pathology reports (the "gold standard" for diagnosis). According to this criterion, 585 NBI images with colorectal polyps were collected from the records of 87 patients. A senior endoscopist classified the 585 NBI images into three categories based on the NICE classification method. During this classification process, the endoscopist's eye movement information was recorded and used to generate the gaze attention images. A patient-level data splitting was then performed to develop and validate the proposed methods. The selected patients and their corresponding images were randomly split into three sets, of which 60% was used for training, 20% was used for validation, and the remaining 20% was used for testing. The training and validation sets included the original image and the gaze attention images generated from eye movement. In the test set, only the original NBI images were included to test the actual performance of the trained models. The detailed information on the dataset is shown in Table 2.

## Experimental setup

A CNN is a complex model with many parameters and requires a large amount of data to obtain more applicable results. With the limited amount of annotated training data, image augmentation was adopted to increase the diversity of training data and improve the generalization ability of deep learning models. Before training, the images were preprocessed and enhanced by randomly applying various rotational transformations, randomly flipping left and right, up and down.

To ensure efficient training and performance optimization of the model in the colorectal polyp classification tasks, this study designed the key parameters of the experiment to meet the special requirements of medical image classification tasks. The backbone of the classification model was pre-trained using ImageNet and then fine-tuned on the experiment's dataset.

The optimizer used for this study was Adam (Adaptive Moment Estimation), which is an adaptive optimization algorithm that combines the advantages of momentum (a method to speed up training) and RMSProp. Adam can dynamically adjust the learning rate of each parameter, quickly approaching the optimal solution with a larger step size in the early stages of training and finely optimizing the parameters with smaller steps in the later stages of training. This feature is particularly suitable for medical image classification tasks with high-dimensional feature spaces and limited data samples. In addition, the Adam optimizer is less sensitive to hyperparameter settings and can provide a more stable

convergence performance during model training. In order to further improve optimization efficiency, this study set the initial learning rate to 0.001, which is based on commonly used empirical values in medical classification tasks, and the optimizer's best performance was verified through multiple experiments (*Zou et al., 2021*).

The learning rate scheduling strategy adopted cosine annealing warm restarts, which can dynamically adjust the learning rate at different stages of training to balance the requirements of fast convergence and meticulous optimization. Specifically, this study used the CosineAnnealingWarmRestarts scheduler in PyTorch (*Loshchilov & Hutter, 2016*), with an initial annealing period set to 10 epochs ($T_0 = 10$) and subsequent periods increasing exponentially ($T_{mult} = 2$). In each cycle, the learning rate gradually decreases from high to low according to the cosine function, thereby promoting the model to quickly jump out of local optima, preventing premature entry into local minima, and improving the robustness of the model. At the end of the cycle, a restart mechanism is used to restore the higher learning rate and explore new solution spaces.

For data processing, the training batch size was set to 8 and the validation batch size was set to 16 in order to balance GPU memory and training efficiency. The training batch size was as large as possible to improve computational efficiency. Due to the limitations of device memory, this selected training batch size could avoid memory overflow and provide more frequent gradient updates to accelerate optimization. During the verification phase, there is no requirement to calculate gradients, therefore larger batch sizes could be used to improve verification efficiency. Data augmentation (random flipping and rotation) was simultaneously used to increase sample diversity and reduce the risk of overfitting. This study set the maximum training epoch to 150 and dynamically adjusted the model preservation strategy based on the performance of the validation set. After each round of training, the classification accuracy of the model on the validation set was evaluated. If the validation accuracy of the current model was better than the historical best value, the weight parameters of the model were saved to ensure the optimal performance of the final model on the validation set. All experiments were run on a 12 GB NVIDIA RTX 4070Ti GPU, which has high-performance computing capabilities and can efficiently support the training needs of high-resolution medical images.

## Evaluation metrics

Quantitative evaluation was conducted by comparing the model's best test accuracy (Acc), precision (Pre), recall (Rec), F1-score (F1 Score), Matthews correlation coefficient (MCC), and Cohen's kappa (Kappa). The definitions of these evaluation metrics are as follows:

$$Acc = \frac{TP + TN}{TP + FP + TN + FN} \tag{7}$$

$$Pre = \frac{TP}{TP + FP} \tag{8}$$

$$Rec = \frac{TP}{TP + FN} \tag{9}$$

$$F1\ Score = 2 \times \frac{Precision \times Recall}{Precision + Recall} \tag{10}$$

$$\text{MCC} = \frac{TP \cdot TN - FP \cdot FN}{\sqrt{(TP + FP)(TP + FN)(TN + FP)(TN + FN)}} \qquad (11)$$

$$Kappa = \frac{p_0 - p_e}{1 - p_e} \qquad (12)$$

TP, FP, TN, and FN represent true positive, false positive, true negative, and false negative, respectively. $P_0$ represents the observed consistency (*i.e.*, the probability of a correct prediction). $P_e$ represents random consistency (*i.e.,* assuming the probability of consistency when the model makes random predictions). As this is a multi-class classification task, the "one-vs-rest" method was used to calculate the performance parameters of each class, treating the target class as a positive class and the remaining classes as negative classes.

## RESULTS

This study conducted experiments using a lightweight EfficientNet_b0 model to validate the effectiveness of the proposed loss function (a combination of CE and MSE). The classification performance of models trained with our proposed loss function was compared to those trained with solely CE or MSE. The comparison results, shown in Table 3, demonstrate that the model achieved optimal performance across all evaluation metrics when using the proposed loss function.

Six different popular CNN architectures (ResNet-18, ResNet-50, ResNeXt-101, EfficientNet_B0, EfficientNet_B1, and EfficientNet_B2) were used as the backbone for feature extraction to validate the proposed method. Each of these backbones has a wide range of applications in the medical image analysis field. The results of enabling and disabling gaze attention in classification backbone training were compared across the six backbones. For example, EfficientNet_b0 referred to the model with EfficientNet_b0 as the backbone, while EfficientNet_b0+Gaze indicated that additional gaze attention maps were added to each image. According to the EfficientNet_b0 experiment, the training time of the proposed method was 21.6 min, and the inference time for a single image was 34.8 milliseconds. The training and inference of this model requires at least 7 GB of GPU memory to be efficiently completed, and the model can meet the needs of practical applications. The computational cost of the model is relatively small, making it suitable for deployment on hardware with limited resources.

As seen in Table 4, most evaluation metrics showed a continuous improvement trend with the additional supervision of gaze attention, which verified the hypothesis that introducing gaze information could improve classification performance. Among all models, EfficientNet_b1+Gaze achieved the best performance and the Acc, Rec, F1-score, MCC, and Kappa values were 0.8696, 0.8841, 0.8816, 0.6999, and 0.6998 respectively. Although EfficientNet_b0+Gaze had the highest Pre score of 0.8815, the difference was minimal with EfficientNet_b1+Gaze scoring 0.8792. Based on other evaluation indicators, EfficientNet_b1+Gaze demonstrated the best overall performance in classifying colorectal polyps. Figure 4 presents the confusion matrices for EfficientNet_b1: (Fig. 4A) without gaze attention and (Fig. 4B) with gaze attention. The confusion matrices can clearly show

**Table 3 Comparison of different loss functions.**

|  | Acc | Pre | Rec | F1 | MCC | Kappa |
|---|---|---|---|---|---|---|
| MSE | 0.6521 | 0.2404 | 0.3049 | 0.2688 | 0.0116 | 0.009 |
| CE | 0.8609 | 0.8745 | 0.8644 | 0.8691 | 0.6711 | 0.6705 |
| MSE+CE | **0.8696** | **0.8815** | **0.8763** | **0.8788** | **0.6941** | **0.6940** |

Note:
Remark: Boldface number means the best for each metric or score. EfficientNet_b0 as the base model.

**Table 4 Ablation study of different CNNs as a backbone with and without gaze attention.**

| Methods | Acc | Pre | Rec | F1 | MCC | Kappa |
|---|---|---|---|---|---|---|
| EfficientNet_b0 | 0.8087 | 0.7731 | 0.8322 | 0.7998 | 0.5704 | 0.5695 |
| EfficientNet_b0+Gaze | 0.8696 | **0.8815** | 0.8763 | 0.8788 | 0.6941 | 0.6940 |
| EfficientNet_b1 | 0.8607 | 0.8233 | 0.88 | 0.8491 | 0.6845 | 0.684 |
| EfficientNet_b1+Gaze | **0.8696** | 0.8792 | **0.8841** | **0.8816** | **0.6999** | **0.6998** |
| EfficientNet_b2 | 0.8261 | 0.8403 | 0.8481 | 0.8439 | 0.6039 | 0.6034 |
| EfficientNet_b2+Gaze | 0.8609 | 0.8745 | 0.8644 | 0.8691 | 0.6711 | 0.6705 |
| ResNet18 | 0.8521 | 0.8149 | 0.876 | 0.8425 | 0.6684 | 0.6673 |
| ResNet18+Gaze | 0.8347 | 0.8487 | 0.8444 | 0.8465 | 0.6126 | 0.6125 |
| ResNet50 | 0.8 | 0.8159 | 0.7967 | 0.8043 | 0.5153 | 0.5121 |
| ResNet50+Gaze | 0.8087 | 0.8246 | 0.8087 | 0.8153 | 0.5153 | 0.5122 |
| ResNeXt-101-32x8d | 0.7913 | 0.8068 | 0.7848 | 0.7929 | 0.4902 | 0.4858 |
| ResNeXt-101-32x8d +Gaze | 0.8522 | 0.8757 | 0.8368 | 0.8512 | 0.64 | 0.6321 |

Note:
Remark: Boldface number means the best for each metric or score. Acc indicates accuracy, Pre indicates precision, Rec indicates recall, F1 indicates F1-score, MCC indicates Matthews Correlation Coefficient, Kappa indicates Cohen's Kappa.

how the models are confused when they make predictions. The columns of the matrices represent the true labels, and the rows denote the predicted labels. The figure shows that the model can easily distinguish Type 3 polyps from other categories regardless of if the eye movement information was added. However, the differences in the characteristics of Type 1 and Type 2 polyps are minor, and it is easy to misclassify between them without adding eye movement information. After adding eye movement information, the error rate of classification was reduced, achieving a higher accuracy rate. The above results demonstrated the crucial role of gaze attention in classification performance enhancement. The significance of adding gaze attention was also verified using the average area under the receiver operating characteristic (ROC) curve (AUC). The AUC value ranges from 0.5 to 1, with a value closer to 1 indicating an excellent classifier. Figure 5A illustrates the ROC curves for EfficientNet_b1, while Fig. 5B presents the ROC curves for EfficientNet_b1 combined with Gaze. The ROC curve for EfficientNet_b1+Gaze exhibited a steeper upward trend, with an AUC value of 0.9022, highlighting its superior performance. This indicated that the improved model can control the misjudgment rate at a lower level while maintaining a high recall rate.

Figure 6 shows the effect of enabling or disabling gaze attention. When gaze attention was enabled, the model focused more on regions of interest when analyzing an image,

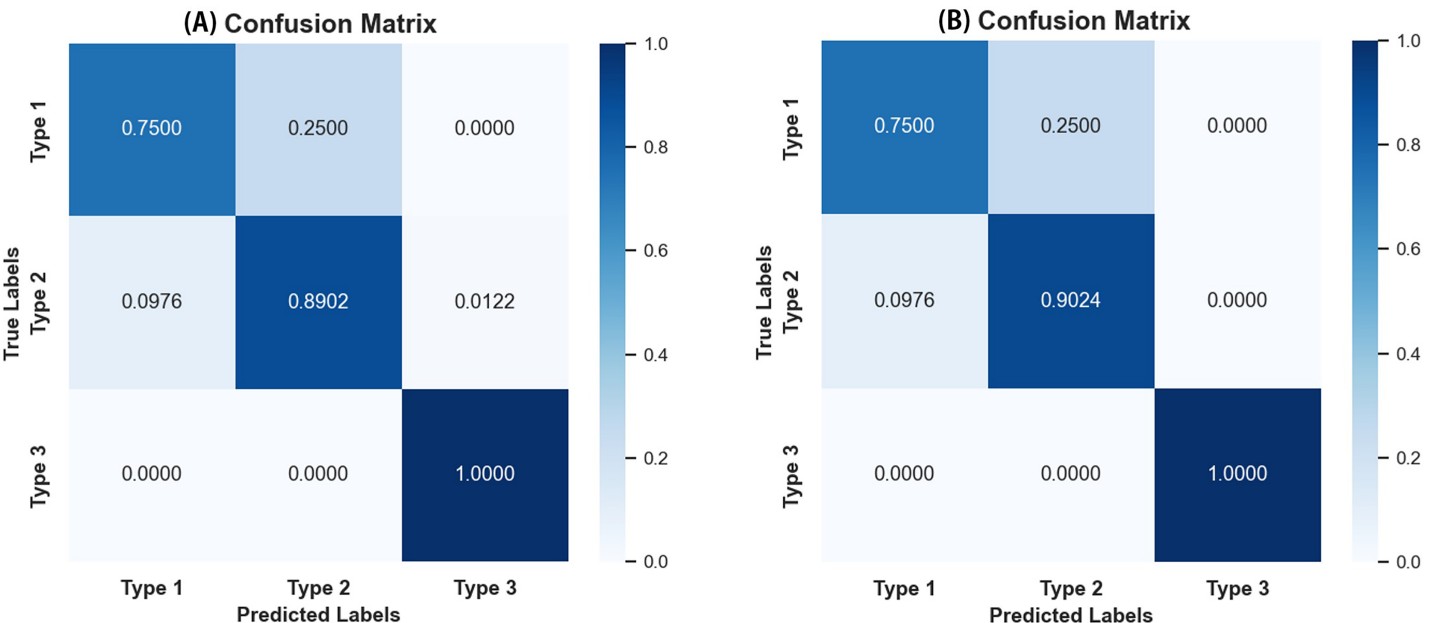

**Figure 4** **Confusion matrices of EfficientNet_b1.** (A) Confusion matrix without gaze attention enabled; (B) confusion matrix with gaze attention enabled.                                     

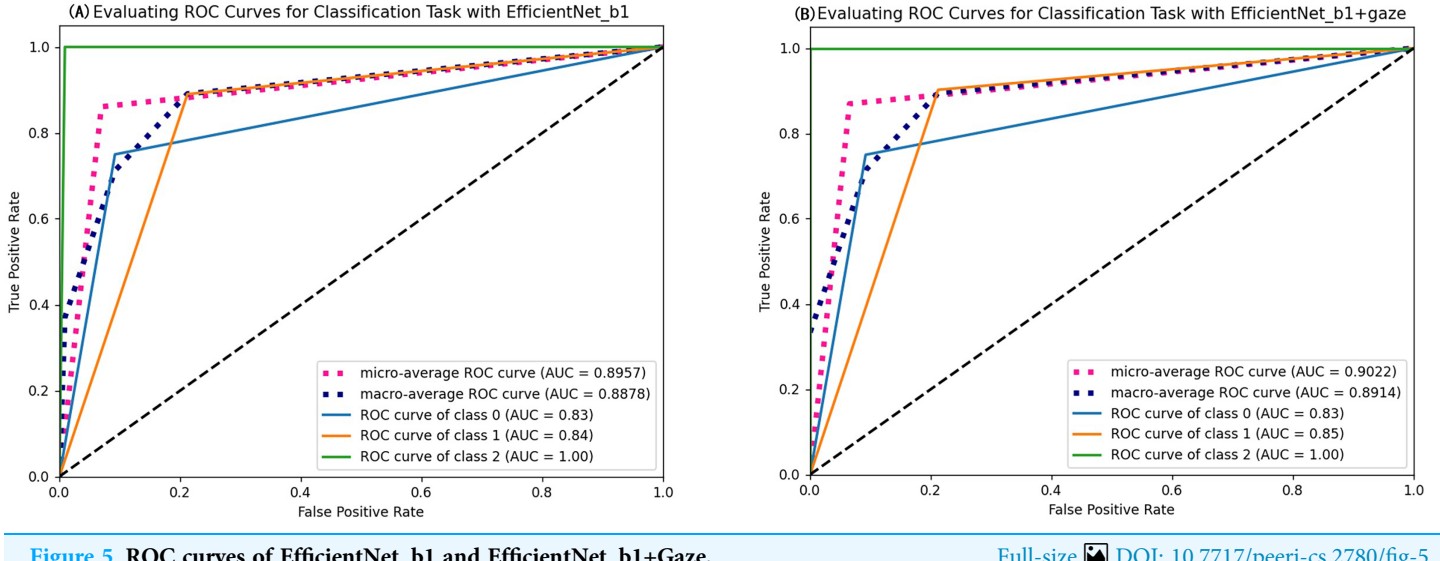

**Figure 5** **ROC curves of EfficientNet_b1 and EfficientNet_b1+Gaze.**                     

which were usually determined automatically by the model and were often key features relevant to the task. In the second column, the effect was shown using CAM, a technique commonly used to visualize regions of interest for deep learning models. Although CAM can help to identify the regions of concern of the model, it does not guarantee that these regions will be the same as those determined by a human doctor. Instead, as seen in the third column, the gaze-attention mechanism was added, which enabled the model to learn and focus on important task-related regions more autonomously. This mechanism mimicked human visual attention and allowed the model to focus more on important

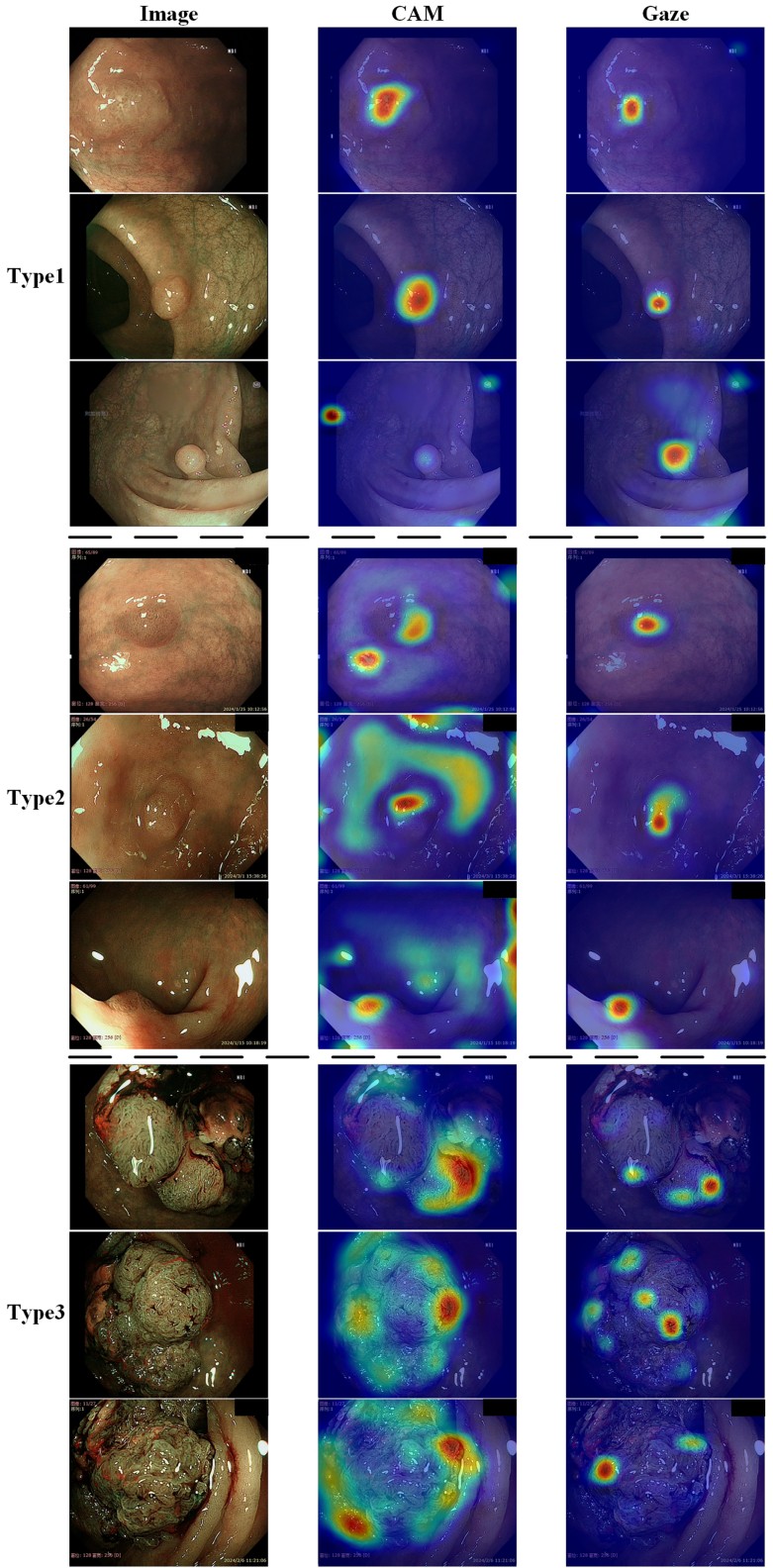

**Figure 6 Demonstration of the effects of enabling or disabling gaze attention.** The first category shows pictures of the test; the second column shows the effect of using CAM; and the third column shows the effect of adding gaze attention.          

features. As a result, with gaze attention enabled, the model focused more accurately on the lesion site in the image, which helped to improve the accuracy and performance of the classification task. The advantage of this approach is that it enabled the model to choose the region of attention more intelligently, rather than simply relying on general activation mapping. In this way, the model was able to better understand the context and key features in the image and thus perform classification and recognition tasks more accurately. Analyzing the experimental results showed that, compared to the original model, the improved model had stronger robustness in handling complex sample situations and improved performance in identifying boundary conditions. This indicated that the study's improved algorithm was more effective in capturing key features in the image, thereby improving the generalization ability and accuracy of the classifier.

## CONCLUSIONS

This study introduced a gaze-based attention network for colorectal polyp classification. A colorectal polyp gaze dataset was constructed that included three types of colorectal polyps and the corresponding gaze images. Then, a CNN model that incorporated endoscopists' gaze information was designed to assist in the classification of colorectal polyps. The gaze information served as a compensation signal to be combined with the original image to train the network. This approach ensured that the model accurately located the colorectal polyps without manual marking by the endoscopists. After completing the training process, the model could directly classify images without relying on eye-tracker data. This method not only effectively improved the performance of the model, but also reduced the dependence on large-scale image data, providing a more efficient and sustainable solution to limitations in current colorectal polyp classification methodologies.

This study has several limitations. Firstly, while gaze information can help improve the accuracy of classification networks and avoid additional annotations, it heavily relies on eye-tracking devices. In practical applications, clinical experts must follow established protocols for image collection. Additionally, gaze patterns are subjective and may vary significantly among different experts, as each expert has their own distinct image reading habits. Although experimental results showed that the proposed strategy was effective in terms of performance and improved diagnostic capabilities, the optimal utilization of gaze remains an unresolved issue. Secondly, the constructed gaze dataset was relatively small, and its impact on colorectal polyp classification ability could be further explored by supplementing it with additional training datasets. Finally, to comprehensively validate the effectiveness of the proposed gaze classification system, the use of data sets from other medical modalities could be considered. Future studies plan to use eye-tracking in more challenging modes. In addition to eye gaze information, additional behavioral data could be collected from doctors including pupil size, reaction time, and mouse cursor movements, in order to build a more comprehensive analysis framework.

### Funding

This work was supported by the Science and Technology Development Fund of Macau SAR (No. 0026/2022/A), the Guangdong Basic and Applied Basic Research Fund, Shenzhen Joint Fund (Guangdong-Shenzhen Joint Fund), Guangdong-Hong Kong-Macau Research Team Project (No. 2021B1515130003), the Key Research and Development Plan of Hubei Province (No. 2022BCE034), the Natural Science Foundation of Hubei Province (No. 2024AFB1054), Joint Funds of the Natural Science Foundation of Hubei Province (No. 2022CFD080). The funders had no role in study design, data collection and analysis, decision to publish, or preparation of the manuscript.

### Grant Disclosures

The following grant information was disclosed by the authors:
Science and Technology Development Fund of Macau SAR: 0026/2022/A.
Guangdong Basic and Applied Basic Research Fund, Shenzhen.
Guangdong-Hong Kong-Macau Research Team: 2021B1515130003.
Key Research and Development Plan of Hubei Province: 2022BCE034.
Natural Science Foundation of Hubei Province: 2024AFB1054 and 2022CFD080.

### Competing Interests

The authors declare that they have no competing interests.

### Author Contributions

- Zhenghao Guo conceived and designed the experiments, performed the experiments, analyzed the data, performed the computation work, prepared figures and/or tables, and approved the final draft.
- Yanyan Hu conceived and designed the experiments, performed the experiments, analyzed the data, authored or reviewed drafts of the article, and approved the final draft.
- Peixuan Ge conceived and designed the experiments, performed the experiments, analyzed the data, performed the computation work, prepared figures and/or tables, authored or reviewed drafts of the article, and approved the final draft.
- In Neng Chan performed the experiments, analyzed the data, authored or reviewed drafts of the article, and approved the final draft.
- Tao Yan conceived and designed the experiments, performed the experiments, analyzed the data, performed the computation work, prepared figures and/or tables, authored or reviewed drafts of the article, and approved the final draft.
- Pak Kin Wong conceived and designed the experiments, analyzed the data, authored or reviewed drafts of the article, and approved the final draft.
- Shaoyong Xu analyzed the data, authored or reviewed drafts of the article, and approved the final draft.

- Zheng Li performed the experiments, analyzed the data, authored or reviewed drafts of the article, and approved the final draft.
- Shan Gao analyzed the data, authored or reviewed drafts of the article, and approved the final draft.

## Ethics

The following information was supplied relating to ethical approvals (*i.e.,* approving body and any reference numbers):

This study was approved by the Institutional Review Board of Xiangyang Central Hospital (IRB approval number: 2024-145).

## Data Availability

The data is available at Zenodo: T. (2024). Colorectal-polyps-gaze-dataset [Data set]. Zenodo. https://doi.org/10.5281/zenodo.13824600.

## Supplemental Information

Supplemental information for this article can be found online at http://dx.doi.org/10.7717/peerj-cs.2780#supplemental-information.

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
