# Peer review of "Enhancing colorectal polyp classification using gaze-based attention networks"

_PeerJ Computer Science, doi:10.7717/peerj-cs.2780_

## Round 0.1 · original submission · Major Revisions

As per the reviewer comments I am asking that you revise the manuscript.

Reviewer 1 ·

Basic reporting

All comments are included in detail in the fourth section.

Experimental design

All comments are included in detail in the fourth section.

Validity of the findings

All comments are included in detail in the fourth section.

Additional comments

Review Report for PeerJ Computer Science
(Enhancing colorectal polyp classification using Gaze-based Attention Networks)

1. Within the scope of the study, various multi-class classification operations were performed on the colorectal polyp dataset using deep learning-based approaches.

2. In the introduction section, colorectal cancer ranks, the importance of the subject, and the main contributions of the study were mentioned. It is suggested that the main contributions section be detailed and the model-based originality point be explained more clearly.

3. In the related works section, studies in the literature based on both eye-tracking technology and studies on colorectal polyp classification using deep learning models based on convolutional neural networks were mentioned. Although the basic literature was mentioned in this section, a literature table consisting of columns such as "used/proposed model, dataset, classes, results, pros and cons" should be added and the advantages of this study compared to the literature should be expressed more clearly.

4. It was stated that colonoscopy data collected from Xiangyang Central Hospital with the approval of the ethics committee was preferred as the dataset in the study. The collection of the dataset specific to the study by obtaining the necessary ethics committee permission is very valuable in terms of the originality of the dataset, usability and contribution to the literature.

5. It is necessary to specify in detail why the type and/or amount of all necessary parameters such as optimizer, learning rate, epoch in the training phase of the classification model are completely preferred or how they are determined with a justification.

6. It is observed that the content of the proposed model and backbone diversity are of a certain quality. However, it should be clearly explained how the training, validation, testing distribution percentages are determined and the study should be interpreted in terms of cross-validation.

7. There are deficiencies in the evaluation metrics section. For the correct analysis of the classification results, the metrics must be obtained completely. For this reason, it is recommended to obtain metrics such as Cohen's Kappa score and Matthews Correlation Coefficient.

As a result, although the study has the potential to contribute to the literature regarding the classification of artificial intelligence colorectal polyps, all the sections listed above should be addressed.

Cite this review as

Reviewer 2 ·

Basic reporting

- There are some grammatical errors in the text, it should be revised.
- Symbols and terms should be written in italics.
- The sentence "Although NICE classification has high diagnostic accuracy for the type of polyp, However, due to its subjective nature, inter-observer agreement varies from case to case, especially for non-experts" should be edited.
- Information about colorectal polyps should be given in the first sentences of the abstract, even if there is a sentence.
- The steps of the proposed method for the classification of colorectal polyps should be briefly presented in the abstract.
- The contributions mentioned in the introduction should be detailed. Each one should explain what purpose the study was designed to achieve.
-A paragraph containing the organization of the paper should be added at the end of the introduction.

Experimental design

- What is the computational cost of the proposed method?
- Why was MSE used?
- The captions for images A and B in Figure 4 are the same, what is the difference?

Validity of the findings

- An example can be given for the Gaze attention map.
- Explain what the 3 nodes mentioned are, what is their relationship with the output?

Additional comments

I have examined the paper in detail. When the experimental results are examined, I see that the analysis of the results is only done in a table. I think that more detailed experiments should be done and the analyses should be more comprehensive in order to publish articles in SCI journals. The results are given only for the authors' own methods. The contribution of the method to the literature should be shown with the experimental results by making comparisons with different studies in the literature. Although the subject of interest is hot and important, I think that the contribution of the paper to the literature is weak in this state.

Although I have written the minor corrections that should be made in the paper, my opinion about the paper is negative.

Cite this review as

---

## Round 0.2 · accepted · Accept

The author has addressed the reviewers' comments properly, so I think the manuscript can be published.

Reviewer 1 ·

Basic reporting

All comments are included in detail in the fourth section.

Experimental design

All comments are included in detail in the fourth section.

Validity of the findings

All comments are included in detail in the fourth section.

Additional comments

Review Report for PeerJ Computer Science
(Enhancing colorectal polyp classification using Gaze-based Attention Networks)

Thanks for the revision. Changes in the paper based on reviewer comments are sufficient. The paper has a high potential to contribute to the literature. Best regards.

Cite this review as

Reviewer 2 ·

Basic reporting

No comment

Experimental design

No comment

Validity of the findings

No comment

Additional comments

The authors have revised the paper, taking into account all suggestions. It can be accepted as it is.

Cite this review as